# New Cobalt (II) Complexes with Imidazole Derivatives: Antimicrobial Efficiency against Planktonic and Adherent Microbes and In Vitro Cytotoxicity Features

**DOI:** 10.3390/molecules26010055

**Published:** 2020-12-24

**Authors:** Alina Fudulu, Rodica Olar, Cătălin Maxim, Gina Vasile Scăeţeanu, Coralia Bleotu, Lilia Matei, Mariana Carmen Chifiriuc, Mihaela Badea

**Affiliations:** 1Stefan S. Nicolau Institute of Virology, Romanian Academy, 030304 Bucharest, Romania; alina.fudulu@virology.ro (A.F.); coralia.bleotu@virology.ro (C.B.); lilia.matei@virology.ro (L.M.); 2Department of Inorganic Chemistry, Faculty of Chemistry, University of Bucharest, 90–92 Panduri Str., 050663 Bucharest, Romania; rodica.olar@chimie.unibuc.ro (R.O.); catalin.maxim@chimie.unibuc.ro (C.M.); 3Department of Soil Sciences, University of Agronomical Sciences and Veterinary Medicine, 59 Mărăști Str., 011464 Bucharest, Romania; gina.scaeteanu@agro-bucuresti.ro; 4Department of Microbiology, Faculty of Biology, University of Bucharest, 1–3 Aleea Portocalelor Str., 60101 Bucharest, Romania; carmen.chifiriuc@bio.unibuc.ro; 5Academy of Romanian Scientists, 010071 Bucharest, Romania

**Keywords:** cobalt complexes, imidazole derivatives, carboxilato ligand, antimicrobial activity, cytotoxicity

## Abstract

Three novel Co(II) complexes of the type [Co(C_4_H_5_O_2_)_2_L_2_] (where C_4_H_5_O_2_ is methacrylate anion; L = C_3_H_4_N_2_ (imidazole; HIm) (**1**), C_4_H_6_N_2_ (2-methylimidazole; 2-MeIm) (**2**), C_5_H_8_N_2_ (2-ethylimidazole; 2-EtIm) (**3**)) have been synthesized and characterized by elemental analysis, IR and UV-Vis spectroscopic techniques, thermal analysis and single crystal X-ray diffraction. X-ray crystallography revealed for complexes (**1**) and (**2**) distorted trigonal bipyramid stereochemistry for Co(II), meanwhile for complex (**3**) evidenced that the unit cell comprises three molecular units with interesting structural features. In each unit, both stereochemistry adopted by metallic ion and coordination modes of carboxylate anions are different. The screening of antimicrobial activity revealed that *Candida albicans* planktonic cells were the most susceptible, with minimal inhibitory concentration (MIC) values of 7.8 μg/mL for complexes (**1**) and (**2**) and 15.6 μg/mL for complex (**3**). Complexes (**1**) and (**2**) proved to be more active than complex (**3**) against the tested bacterial strains, both in planktonic and biofilm growth state, with MIC and minimal biofilm eradication concentration (MBEC) values ranging from 15.6 to 62.5 μg/mL, the best antibacterial effects being noticed against *Staphylococcus aureus* and *Pseudomonas aeruginosa*. Remarkably, the MBEC values obtained for the four tested bacterial strains were either identical or even lower than the MIC ones. The cytotoxicity assay indicated that the tested complexes affected the cellular cycle of HeLa, HCT-8, and MG63 cells, probably by inhibiting the expression of vimentin and transient receptor potential canonical 1 (TRPC1). The obtained biological results recommend these complexes as potential candidates for the development of novel anti-biofilm agents.

## 1. Introduction

The coordination compounds containing both carboxylate and azole type ligands are interesting for at least two main reasons: (**i**) the versatility of carboxylate anions that could easily form complexes with interesting structural features and special properties; (**ii**) the imidazole ring acts as the coordination site in many metalloproteins (e.g., the active site of the metalloenzyme copper-zinc superoxide dismutase contains imidazolato ligand of histidine residues); therefore, metal complexes containing imidazole derivatives are most likely to exhibit a biological activity and (**iii**) the proven antimicrobial, antidiabetic, anticancer activities of azole ligands ensure a promising potential of complexes for the development of new drugs [1,2,3].

An interesting aspect regarding synthesis of complexes with carboxylate anions and imidazole derivatives results in simultaneous formation of more compounds from the same system. Thus, from the reaction of copper(II) acrylate/methacrylate with imidazole (HIm), two different types of complexes have been obtained: mononuclear square planar [Cu(RCOO)_2_(Him)_2_] and binuclear [Cu_2_(RCOO)_4_(Him)_2_] species [4]. It has been observed that the mononuclear compound undergoes interconversion in the binuclear species in methanol solution, but not in anhydrous methanol. Additionally, the mononuclear complex containing methacrylate was reported as first mononuclear square planar copper (II) complex with unidentate carboxylato ligand [4]. From the same raw materials but using different molar ratio, trinuclear complexes with general formula Cu_3_A_5_(OH)(HIm)_3_, where A is acrylate/methacrylate, were obtained [5].

Another study [6] revealed that the reaction between copper (II) acrylate and 2-ethylimidazole produced geometric isomers of bis(acrylato)-bis-(2-ethylimidazole)-copper(II) of which *cis* isomer was more active than the *trans* one against Gram positive and fungal strains.

For complexes of type Zn(HIm)_2_(L_1_)_2_, Zn(HIm)_2_(L_2_)_2_ and [Cd(HIm)_2_(L_2_)_2_]·4H_2_O (L_1_=3-(4-methoxyphenyl)acrylate; L_2_ = 3,4-(methylenedioxy)benzoate), on the basis of X-ray structure, it was evidenced that imidazole units act as unidentate ligands through their iminic nitrogen atom. Furthermore, rich intra- and intermolecular noncovalent interactions (C-H···O, CH_2_···O, CH_3_···O, C-H···π, CH_3_-π, O-H···π, π-π interactions) led to different structures (3D network, 3D prismatic layer network, 3D layer network structure) [7]. Similarly, in the case of dimer Cu_2_(Ac)_4_(2-EtIm)_2_ (Ac = acetate; 2-EtIm = 2-ethylimidazole), the solid 3D framework and stability were sustained by the presence of a strong intermolecular hydrogen bond between neighboring dimers through the pyrrolic nitrogen in the imidazole derivative and an acetate group [8].

With respect to biological properties of imidazole containing complexes, copper (II) acetate imidazole has been reported to have anticonvulsant [9,10] and hypoglycaemic [11] activities.

Copper(II) carboxylate complexes with imidazole/imidazole derivatives exhibited SOD-like activity [3] and antitumor effects [12,13] and are considered DNA binding models [14,15].

Furthermore, Tabrizi and co-workers [16] reported that complexes [Cd(valp)_2_(HIm)_2_], [Co(valp)_2_(HIm)_2_] (valp = valproate ion) exhibit cytotoxic activity against a number of tumor cell lines and present antibacterial activity against Gram positive and Gram negative bacteria.

In addition, complexes of type [AgL_2_(9-aca)] (L = 2-methylimidazole/4-phenylimidazole/2-methylbenzimidazole; 9-aca = 9-anthracenecarboxylic acid) exhibit selective antimicrobial activity against *Candida albicans* [17]. Additionally, the screening of mononuclear, polyinuclear, and polymeric Ag(I) complexes comprising 9-anthracenecarboxylic acid and imidazole and N1-substituted imidazoles (1-methylimidazole, 1-butylimidazole, 1-(3-aminopropyl)imidazole), respectively, evidenced that the complexes were highly active against *Candida albicans, Escherichia coli*, and *Staphylococcus aureus* strains [2].

As a continuation of our previous work on metal carboxylates with azole type ligands [6,18,19,20,21], we report herein the synthesis, structural characterization, and results related to biological activity for three novel complexes of the type [Co(C_4_H_5_O_2_)_2_L_2_] (C_4_H_5_O_2_ = methacrylate anion (Macr); L = C_3_H_4_N_2_ (imidazole; Im) (**1**), C_4_H_6_N_2_ (2-methylimidazole; 2-MeIm) (**2**), C_5_H_8_N_2_ (2-ethylimidazole; 2-EtIm) (**3**).

## 2. Results and Discussion

Three novel complexes of the type [Co(Macr)_2_L_2_] (Macr: methacrylate anion; L: imidazole (HIm) (**1**); 2-methylimidazole (2-MeIm) (**2**); 2-ethylimidazole (2-EtIm) (**3**)) were obtained and fully characterized, through single crystal X-ray diffraction, as mononuclear species.

### 2.1. Complexes Synthesis

The novel complexes were synthesized from cobalt carbonate, methacrylic acid, and imidazole derivatives as raw materials. In a first step, cobalt methacrylate was obtained in methanolic solution from cobalt carbonate and methacrylic acid reaction. To the fresh cobalt methacrylate solution, each imidazole derivative was added, in respect with the molar ratio 1Co:1L. After slow evaporation, single crystals of complexes (**1**) ÷ (**3**) have been obtained.

### 2.2. Complexes Characterization

#### 2.2.1. Description of the X-ray Crystal Structures of the Complexes

The crystal structure for the three complexes was solved by single crystal X ray diffraction and a summary of crystallographic data is presented in Table 1.

In the following, since both compounds (**1**) (Figure 1a) and (**2**) (Figure 1b) present similar structures, we will discuss the two complexes together. The crystal structures reveal the presence of the neutral complexes with two imidazole ligands and two methacrylate anions. The cobalt ion is pentacoordinated with a trigonal bipyramidal stereochemistry, with a continuous shape measure (CShM) value of 3.318 for (**1**) and 3.020 for (**2**) (Appendix A). The trigonal plane is obtained by two nitrogen atoms from two imidazole ligands, and one oxygen atoms from chelating carboxylato ligand. The axial positions are occupied by two oxygen atoms from two different methacrylato ligands (Co1–O1 = 1.982(3); Co1–O4 = 2.457 Å; for (**1**) Co1–O5 = 1.982(3); Co1–O3 = 2.512 Å for (**2**)).

Selected bond distances and angles for the two compounds are collected in Table 2 and Appendix A. The mononuclear entities interact through hydrogen bonds established between the nitrogen atoms of one ligand groups and the oxygen atoms belonging to a neighboring unit resulting in a supramolecular square grid topology (Figure 2) [O2′-N2= 2.687 Å, O3-N4′′= 2.762 Å, ′ = 0.5 + *x*, −0.5 − *y*, *z*, ′′ = −0.5 + *x*, 0.5 − *y*, *z* for (**1**) and O2-N3′ = 2.723 Å, O1-N4′′ = 2.762 Å, ′ = *x*, 1.5 − *y*, 0.5 + *z*, ′′ = *x*, −1 + *y*, *z* for (**2**)].

The structure of complex (**3**) reveals very interesting features. The asymmetric unit contains three crystallographically independent neutral molecules (Figure 3). All three cobalt (II) ions present different stereochemistry. For the Co1 moiety (Figure 3a), the metal centers present a slightly distorted octahedral geometry (with a continuous shape measure (CShM) value of 4.345). The Co(II) ion presents an N_2_O_4_ environment resulting from four coordinated oxygens atoms and two nitrogen atoms from the 2-ethylimidazole ligand. The Co-O distances vary between 2.047(3) and 2.383(3) Å.

In the case of the Co2 ion, we observed the presence of tetrahedral stereochemistry geometry (with a continuous shape measure (CShM) value of 0.905). For this unit, the two carboxylate anions coordinate to the metal ion in a unidentate manner. The angles for the Co(II) tetrahedron vary between 100.76(13)° and 110.79(14)°.

The stereochemistry around Co3 (Figure 3c) is different from compounds (**1**) and (**2**). For this cobalt (II) ion, the stereochemistry is distorted square pyramidal (with a continuous shape measure (CShM) value of 3.225). The coordination number five is completed by two nitrogen atoms arising from the 2-EtIm ligands and three oxygens atoms from one unidentate and one chelating methacrylate anions.

In the network, the mononuclear species are interconnected into a 3D network through H bonds, as shown in Figure 4. The presence of three different tectons affords complex supramolecular H-bonds interactions.

#### 2.2.2. Infrared Spectra

The IR spectra of all complexes, together with the spectra of imidazole derivatives and sodium methacrylate, were recorded in order to obtain information on the coordination modes and ligand donor atoms. Table 3 summarizes the absorption maxima and assignments from these spectra.

All complexes spectra contain bands assigned to imidazole derivatives [22]. As an observation, the band due to NH stretching vibrations (3126–3153 cm^−1^) is shifted towards higher wave numbers, compared to the spectra of free ligands, as a consequence of the involvement in H bonds of this moiety.

The stretching vibrations characteristic of carboxylate groups is shifted and/or split, being in accordance with both unidentate and bidentate coordination modes in all complexes [23,24]. Thus, the characteristic bands for symmetrical and asymmetrical stretching vibrations of carboxylate group appear at 1367 and 1556 cm^−1^ in sodium methacrylate spectrum with their difference (Δ = ν_as_(COO) − ν_s_(COO)) of 189 cm^−1^. For complexes (**1**) and (**2**), the Δ value of 193 cm^−1^ arises from unidentate coordination mode, while that of 172/168 cm^−1^ can be assigned to chelate behavior of methacrylate ion. A value of 200 cm^−1^ of this difference for complex (**3**) accounts for unidentate behavior, while that of 182 cm^−1^ comes from the chelate one.

#### 2.2.3. UV-Vis-NIR Spectra

The electronic spectra of complexes are presented in Figure 5. The characteristic feature of these spectra is the large number of peaks and the high intensity of d–d transitions compared with the ligands bands. Both aspects suggest a lower symmetry and the lack of symmetry center.

For complexes (**1**) and (**2**), the spectra are quite similar, and the d–d transition bands were assigned in accord with a trigonal bipyramidal stereochemistry [25]. For complex (**3**), which contains three different molecules in asymmetric unit, only tetrahedral (T_d_) and square pyramidal stereochemistry can be identified (Table 4).

Taking into account that the biological data were collected using DMSO solutions, the complexes stability in this solvent was investigated by UV-Vis spectroscopy. No major changes were observed in time as an indicative that all compounds are stable for at least 48 h (Appendix A).

#### 2.2.4. Thermal Behavior

Thermal analysis was used in order to obtain information regarding complexes stability in air, presence or absence of solvent molecules and cobalt content. The thermal behavior of the three compounds is quite similar, as shown in Figure 6 for complex (**3**), the data obtained from the TG/DTG/DTA curves being presented in Table 5.

The TG curves suggest that all complexes are stable up to, at least, 220 °C, which is in accordance with the absence of any solvent molecules. Additionally, all compounds melt without decomposition in the temperature range of 119–135 °C.

Thermal decomposition occurs in two stages, consisting in the elimination of imidazole derivatives and the oxidative degradation of the methacrylate anion; the final residue is the mixed oxide Co_3_O_4_.

#### 2.2.5. Antimicrobial Assay

The results of the antimicrobial activity assays against microbial cells in planktonic (Minimal Inhibitory Concentration—MIC values) and biofilm (Minimal Biofilm Eradication Concentration—MBEC values) growth state are summarized in Table 6.

The obtained complexes exhibited an improved antimicrobial activity against the planktonic bacterial strains, as compared with that of the Co(II) methacrylate and ligands. The MIC values obtained for the complexes were, with few exceptions recorded for the two Gram-negative tested strains, from two to 64 times lower than those obtained for the ligands. The exceptions are represented by the *P. aeruginosa* strain, for which the ligands exhibited the same MIC as the complexes and the *E. coli* strain, for which the MIC of complex (**3**) was two times higher than that of the ligand.

The Gram-positive and the fungal strains were significantly more susceptible to the tested complexes, as compared to the ligands. The *C. albicans* fungal strain proved to be the most susceptible microorganism, when grown in planktonic form, the tested complexes being active until very low concentrations, as revealed by the obtained MIC values, of 7.8 μg/mL for complexes (**1**) and (**2**) and 15.6 μg/mL for complex (**3**).

The tested complexes proved to be very active against the biofilms formed by the four tested bacterial strains, the MBEC values being eight to 32 times lower than those of the corresponding ligands. Unfortunately, the fungal biofilm proved to be highly resistant to the entire range of tested concentrations (i.e., from 1000 to 1.9 μg/mL), both in case of ligands and complexes.

Regarding the tested bacterial strains, the most susceptible was *S. aureus,* both in planktonic and biofilm growth state, all tested complexes exhibiting the same MIC and MBEC value of 15.6 μg/mL, followed by *P. aeruginosa,* with MIC values of 31.2–62.5 μg/mL and MBEC values of 15.6–31.2μg/mL.

The comparative analysis of the antimicrobial efficiency of the three complexes revealed that complexes (**1**) and (**2**) were slightly more active than complex (**3**). This could be related to the different stereochemistry around Co3 for compound (**3**), as compared with compounds (**1**) and (**2**), that could influence the affinity of this complex to microbial target cells.

It is remarkable to notice that, although the resistance of microbial biofilms is well known to be much higher in comparison with that of planktonic bacteria, leading to much higher MICs [26], in the case of our compounds, the MIC and MBEC values obtained for the bacterial strains were either identical, or the MBEC vales were even lower than the corresponding MIC ones in three cases, i.e., complexes (**1**) and (**2**) against *P. aeruginosa* biofilms and complex (**3**) against *E. coli* biofilm. These results highlight the potential of these compounds for the development of novel anti-biofilm agents. One of the possible mechanisms of action could be the destabilization of the exopolymeric biofilm matrix, through their Co(II), taking into account that other bivalent cations, such as calcium, magnesium, and iron, have been shown to stabilize the biofilm matrix by enhancing its structural integrity through electrostatic interactions that serve to cross-link the matrix [27].

#### 2.2.6. Cytotoxicity Tests

The influence of the tested compounds on the cellular cycle of three cell lines of different origins, i.e., epithelial (HeLa), colorectal (HCT-8), and osteoblast (MG63) cells, was investigated by flow cytometry (Table 7, Appendix A). The tested compounds induced an evident increase of S and/or G2/M phases, correlated with a decrease of G1 phase.

In order to elucidate the mechanisms responsible for the changes of the cellular cycle of the three cell lines and taking into account that the tested compounds contain Co, which is an inductor of hypoxia, we have further analyzed the mRNA expression levels of some structural molecules related to hypoxia-cell response, i.e., the transient receptor potential canonical 1 (TRPC1), expressed in hypoxia, as well as the cytoskeletal protein vimentin (VIM) and the cytosolic proteins arrest in beta 1 (ARBB1), which are involved in the regulation of the cell cycle (Figure 7). The TRPC1 expression occurs in hypoxia, this molecule being considered as a regulator of specific hypoxia-induced changes and influencing the susceptibility of tumoral cells to conventional therapies [28]. The decreased expression of TRPC1 in the presence of the tested complexes after 24 h of treatment, suggests that the tested compounds do not have the ability to induce hypoxia in the first stage.

The increase in the G2 phase could be explained by the downregulation of vimentin and ARRB1, the prolonged G2 arrest preventing the cells to enter in necrosis or apoptosis [29,30,31].

## 3. Materials and Methods

### 3.1. General Information

High purity reagents were purchased from Merk KGaA (Darmstadt, Germany, methacrylic acid), Fisher Chemical (Waltham, MA, USA, CoCO_3_), and Sigma-Aldrich (Saint-Louis, MO, USA, imidazole derivatives) and were used without further purification.

Chemical analysis of carbon, nitrogen, and hydrogen has been performed using a PE 2400 analyzer (Perkin Elmer, Waltham, MA, USA). IR spectra were recorded in KBr pellets with a Tensor 37 spectrometer (Bruker, Billerica, MA, USA) in the range 400–4000 cm^−1^. UV-Vis-NIR spectra were recorded on solid samples in the range 200–1800 nm, using diffuse reflectance technique and Spectralon as standard, on a V670 spectrophotometer (Jasco, Easton, MD, USA). DMSO solution UV-VIS spectra were recorded on a Jasco V530 spectrophotometer (Jasco, Easton, MD, USA) in the range 250–700 nm. The heating curves (TG and DTA) were recorded using a Labsys 1200 SETARAM instrument, with a sample mass of 11–15 mg over the temperature range of 30–900 °C, using a heating rate of 10 K min^−1^. The measurements were carried out in synthetic air (flow rate 16.66 cm^3^ min^−1^) using alumina crucibles.

X-ray diffraction measurements on single crystal were performed on an IPDS II diffractometer (STOE, Darmstadt, Germany) operating with Mo-Kα (λ = 0.71073 Å) X-ray tube with graphite monochromator. The structure was solved by direct methods and refined by full-matrix least squares techniques based on *F*^2^. The non-H atoms were refined with anisotropic displacement parameters. Calculations were performed using the SHELX-2013 crystallographic software package. A summary of the crystallographic data and the structure refinement are presented in Table 1. Crystallographic data (excluding structure factors) have been deposited with the Cambridge Crystallographic Data Centre with CCDC reference numbers 2034634-2034636. These data can be obtained free of charge via http://www.ccdc.cam.ac.uk/conts/retrieving.html, or from the Cambridge Crystallographic Data Centre, 12 Union Road, Cambridge, CB2 1EZ, UK; fax: (+44) 1223-336-033; or e-mail:deposit@ccdc.cam.ac.uk.

### 3.2. Complexes Synthesis

The mixture formed from 0.48 g CoCO_3_, 0.68 mL methacrylic acid (ρ = 1.02 g/mL), and 40 mL methanol was stirred at room temperature for two hours. The reaction mixture was filtered off to eliminate the unreacted cobalt carbonate. To the purple colored filtrate, a solution containing 20 mL methanol and each imidazole derivative (0.26 g imidazole/0.33 g 2-methylimidazole/0.38 g 2-ethylimidazole) was added. The obtained solutions were stirred at room temperature for two hours and then were allowed to evaporate slowly at room temperature. In time, there were obtained microcrystals colored in different shades of purple depending of imidazole derivative, which were filtered off, washed with methanol, and air dried. Suitable single crystals for X-ray diffraction were obtained by recrystallization from a mixture methanol:DMF (4:1, *v:v*).
(1).[Co(Macr)_2_(HIm)_2_] (purple crystals); soluble in alcohols (methanol, ethanol), dimethylformamide and dimethyl sulfoxide. Anal. Calc.: Co, 16.13; C, 46.04; H, 4.97; N, 15.34; Found: Co, 16.18; C, 46.11; H, 4.95; N, 15.42.(2).[Co(Macr)_2_(2-MeIm)_2_] (blue violet crystals); soluble in alcohols (methanol, ethanol), dimethylformamide and dimethyl sulfoxide. Anal. Calc.: Co, 14.98; C, 48.86; H, 5.64; N, 14.25; Found: Co, 14.92; C, 48.79; H, 5.61; N, 14.33.(3).[Co(Macr)_2_(2-EtIm)_2_] (dark violet crystals); soluble in alcohols (methanol, ethanol), dimethylformamide and dimethyl sulfoxide. Anal. Calc.: Co, 13.98; C, 51.31; H, 6.22; N, 13.30; Found: Co, 13.93; C, 51.37; H, 6.29; N, 13.37.

### 3.3. Microbiological Assays

#### 3.3.1. Quantitative Assay of the Antimicrobial Activity against Planktonic Cells

The liquid medium two-fold serial microdilution method, performed in 96 multi-well plates, allowed the establishment of the MIC values of the tested complexes against Gram positive (*S. aureus* ATCC 6538, *E. feacalis* ATCC 29212) and Gram-negative (*E. coli* ATCC 8739, *P. aeruginosa* ATCC 1671) bacterial and fungal *C. albicans* ATCC 26790 strains. The range of tested concentrations prepared from a stock solution in dimethyl-sulfoxide (DMSO) (10 mg/mL) was from 1000 to 1.9 µg/mL. After incubating the tested compounds with standard microbial suspensions at 0.5 MacFarland density (1.5 × 10^8^ microbial cells/mL) for 24 h at 37 °C, the MIC was established as being the lowest concentration of the compound that inhibited the visible microbial growth in the wells, in comparison with the untreated culture [32,33].

#### 3.3.2. Assessment of the Anti-Biofilm Activity

After the MIC protocol, the same in 96 multi-well plates were further treated to determine the MBEC. In this purpose, the plates were emptied and then washed three times with sterile phosphate buffered saline (PBS) to remove the planktonic cells, and the remaining adherent cells were fixed for 5 min with cold methanol, stained for 20 min with 0.1% aqueous solution of crystal violet, and resuspended in acetic acid 33%. The MBEC corresponded to the lowest concentration that inhibited the microbial adherence, in comparison with the untreated culture [32,33].

### 3.4. Cytotoxicity Tests

#### 3.4.1. Cells

The HeLa (ATCC CCL-2), HCT-8 [HRT-18] (ATCC CCL-244), and MG-63 (ATCC CRL-1427) cell lines have been used. The cells were maintained in Dulbecco’s modified Eagle’s Medium (DMEM):F12, supplemented with 10% foetal bovine serum (FBS), at 37 °C, 5% CO_2_ and humid atmospheres.

#### 3.4.2. Cytotoxicity Assay

The tumor cells treated with 50 and 100 µg/mL cobalt complexes, for 72 h were harvested with trypsin, fixed in cold ethanol (70%), and stained with 50 µg/mL propidium iodide containing 100 µg/mL RNAse A Acquisition was performed using Beckman Coulter flowcytometer (Beckman Coulter, Nyon, Switzerland) and analyses was done using FlowJo software 7.2.5. (Ashland, CA, USA) [21,32].

#### 3.4.3. Semi-Quantitative Real-Time RT-PCR

After treatment with 100 µg/mL compounds for 24 h, total RNA was extracted using TRIzol reagent (Invitrogen, Grand Island, NY, USA) and quantified using a spectrophotometer Eppendorf BioPhotometer Plus (Eppendorf, Framingham, MA, USA). Then, 2 µg of total RNA was reverse transcribed using the High Capacity cDNA Archive Kit (Applied Biosystems, Foster City, CA, USA) according to manufacturer’s indications. The cDNA was subsequently used for quantitative real-time RT-PCR (qRT-PCR) using specific primers for CYP according to [34], and Taqman assay TRPC1 (Hs00608195_m1), VIM (Hs00185584_m1), ARRB1 (Hs00244527_m1) (Applied Biosystems, Waltham, MA, USA). Briefly, 50 ng of cDNA was used as template for each real-time PCR reaction. The amplification was run on StepOne detection system (Applied Biosystems) using Maxima SyBrGreen PCR Master Mix (Applied Biosystems) or Taqman Master Mix (Applied Biosystems, USA) and the following program: 95 °C for 10 min for DNA polymerase heat-activation, followed by 45 cycles comprising denaturation at 95 °C for 15 s, annealing and elongating at 60 °C for 1 min. The relative mRNA expression was presented as 2^−ΔΔCT^, where CT is the cycle threshold.

## 4. Conclusions

New cobalt (II) complexes with mixed ligands (methacrylato and imidazole derivatives) were fully characterized through single crystal X-ray diffraction. Complexes (**1**) and (**2**) exhibit a penta-coordinate stereochemistry with both unidentate and chelate behavior of methacrylato ligand. The complex (**3**) evidenced interesting structural features with three distinctive tectons having differences in both stereochemistry and methacrylato coordination modes. The hydrogen bonds network affords the supramolecular assembling for all species. The three complexes exhibited promising antimicrobial features, being active especially against bacterial biofilms and fungal cells in planktonic growth state.

## Figures and Tables

**Figure 1 molecules-26-00055-f001:**
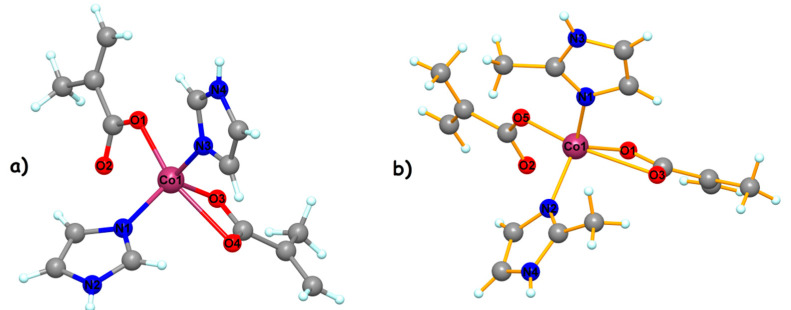
The asymmetric unit for compound (**1**) (**a**) and compound (**2**) (**b**).

**Figure 2 molecules-26-00055-f002:**
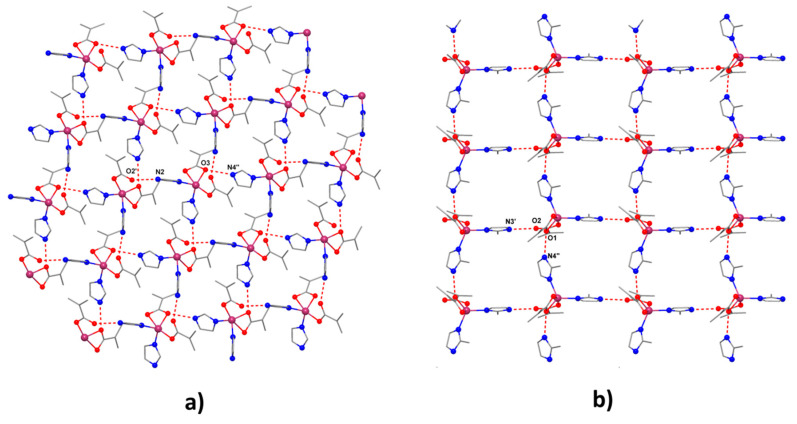
Supramolecular H-bonds networks for (**1**) (**a**) and (**2**) (**b**).

**Figure 3 molecules-26-00055-f003:**
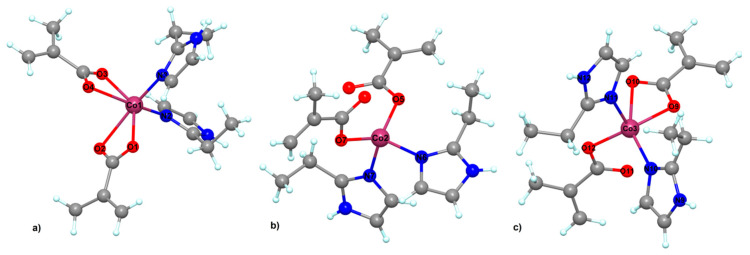
Crystal structure for (**3**) with the presence of a different stereochemistry for cobalt ions: (**a**) octahedral, (**b**) tetrahedral and (**c**) square pyramidal.

**Figure 4 molecules-26-00055-f004:**
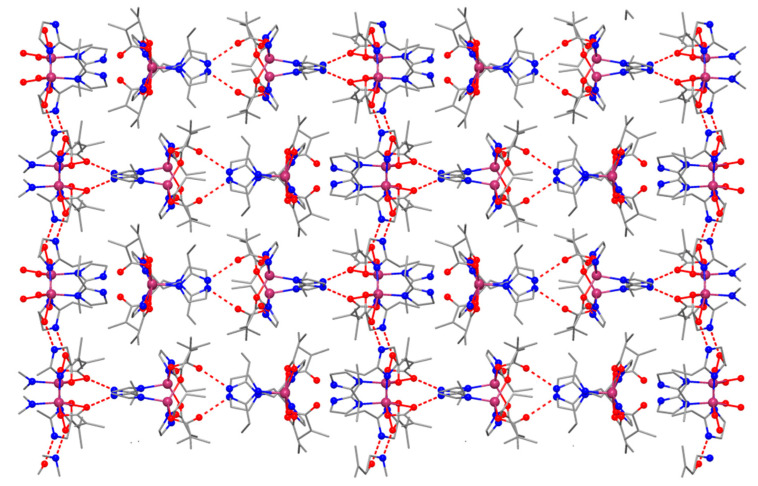
Packing diagram of (**3**) showing the supramolecular H-bonds network.

**Figure 5 molecules-26-00055-f005:**
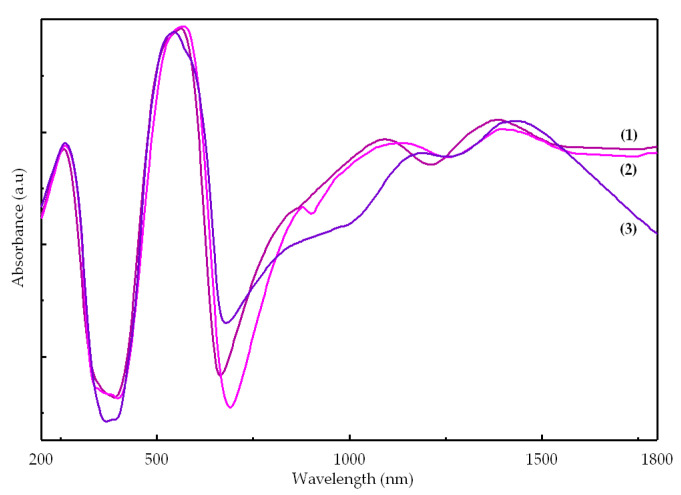
UV-Vis-NIR spectra of complexes.

**Figure 6 molecules-26-00055-f006:**
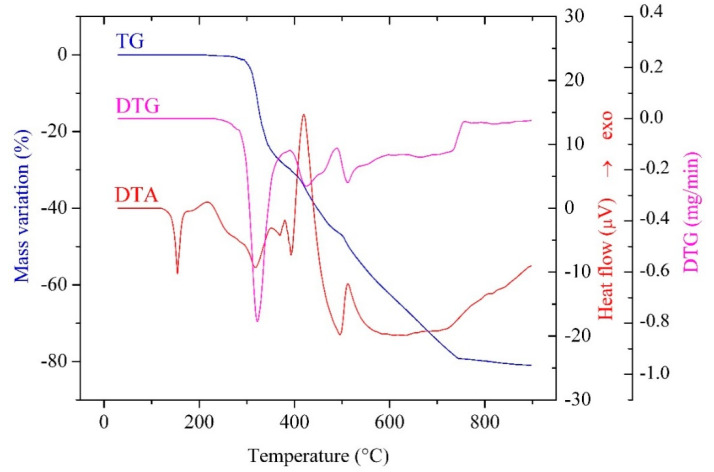
Thermogravimetric (TG), derivative thermogravimetric (DTG), and differential thermal analysis (DTA) curves of [Co(Macr)_2_(2-EtIm)_2_] (**3**) in air flow at 10 °C/min.

**Figure 7 molecules-26-00055-f007:**
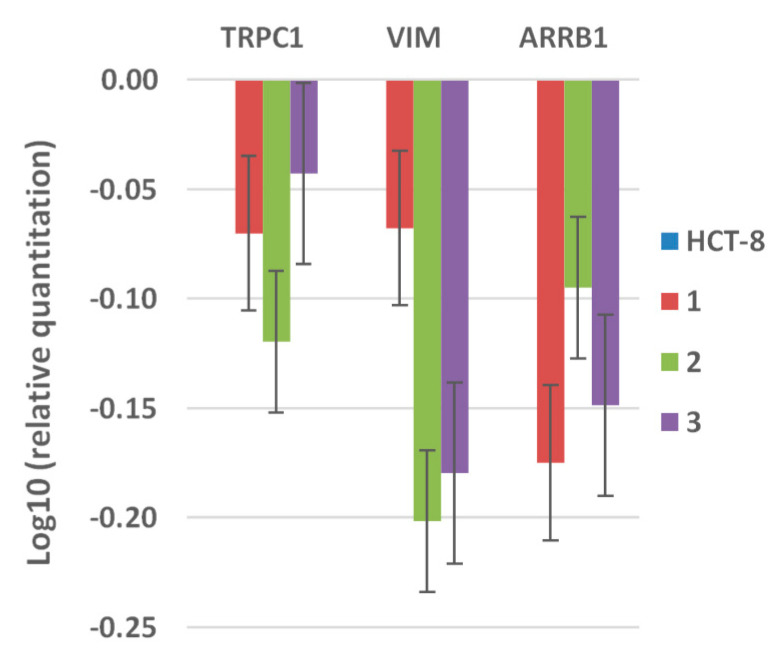
Relative quantification of the expression of TRPC1, VIM, and ARBB1 hypoxia-markers in HCT-8 cell line (expression was normalized with GAPDH gene).

**Table 1 molecules-26-00055-t001:** Crystal data and structure refinement for compounds (**1**)–(**3**).

Compound	(1)	(2)	(3)
Chemical formula	C_14_H_18_CoN_4_O_4_	C_16_H_22_CoN_4_O_4_	C_54_H_78_Co_3_N_12_O_12_
*M* (g·mol^−1^)	365.26	393.31	1264.09
Temperature (K)	293	293	293
Crystal system	monoclinic	monoclinic	monoclinic
Space group	*P*2_1_/a	*P*2_1_/c	*P*2_1_/a
*a* (Å)	12.4662(2)	13.5660(11)	16.2171(2)
*b* (Å)	8.2403(3)	7.8085(4)	16.1886(3)
*c* (Å)	17.4959(4)	18.0920(15)	24.4967(4)
*α* (º)	90	90	90
*β* (º)	104.050(5)	100.707(6)	94.440(5)
*γ* (º)	90	90	90
*V* (Å^3^)	1743.50(9)	1883.1(2)	6411.87(18)
*Z*	4	4	4
*D_c_* (g·cm^−3^)	1.3914	1.3872	1.3094
*µ* (mm^−1^)	1.007	0.938	0.831
*F*(0 0 0)	757.7	821.8	2657.3
Goodness-of-fit (GOF) on *F*^2^	1.033	1.097	0.962
Final *R*_1_, *wR*_2_ [*I > 2σ(l)*]	R1 = 0.0374, wR2 = 0.0869	R1 = 0.0529, wR2 = 0.1241	R1 = 0.0506, wR2 = 0.1135
*R*_1_, *wR*_2_ (all data)	R1 = 0.0730, wR2 = 0.1114	R1 = 0.0848, wR2 = 0.1450	R1 = 0.0974, wR2 = 0.1342
Largest difference in peak and hole (eÅ^−3^)	0.37/−0.42	0.56/−0.64	0.71/−0.94

**Table 2 molecules-26-00055-t002:** Selected bond distances (Å) for (**1**)–(**3**).

(1)			(2)			(3)		
Co(1)	O(1)	1.982(3)	Co(1)	O(1)	1.994(3)	Co(1)	N(3)	2.052(3)
Co(1)	O(3)	2.003(3)	Co(1)	O(5)	1.982(3)	Co(1)	N(2)	2.059(3)
Co(1)	N(3)	2.026(3)	Co(1)	N(2)	2.017(4)	Co(1)	O(1)	2.047(3)
Co(1)	N(1)	2.008(3)	Co(1)	N(1)	2.037(4)	Co(1)	O(3)	2.047(3)
						Co(1)	O(4)	2.352(3)
						Co(1)	O(2)	2.383(3)
						Co(3)	N(10)	2.057(4)
						Co(3)	N(11)	2.020(3)
						Co(3)	O(12)	1.980(3)
						Co(3)	O(9)	2.022(3)
						Co(2)	N(6)	2.028(3)
						Co(2)	N(7)	2.026(3)
						Co(2)	O(5)	1.948(3)
						Co(2)	O(7)	1.958(3)

**Table 3 molecules-26-00055-t003:** Infrared absorption bands (cm^−1^) for ligands and complexes.

NaMacr	HIm	2-MeIm	2-EtIm	(1)	(2)	(3)	Assignments
-	3126 m	3136 m	3153 w	3139 m	3160 m	3175 m	ν(CH), ν(NH)
2969 w	-	-	-	2959 m	2963 m	2976 m	ν_as_(CH_3_)
2928 w	-	-	-	-	2920 m	2927 m	ν_s_(CH_3_)
-	1593 m	1596 vs.	-	1581vs.	1587 vs.	-	δ(NH), ν(CC), ν(CN)
1556 vs.	-	-	-	1561 vs. 1540 s	1561 vs. 1548 s	1568 vs. -	ν_as_(COO)
1367 m	-	-	-	- 1368 m	1380 m 1368 s	1386 m 1368 s	ν_s_(COO)
-	1251 w	1206 w	1243 w	1257 m	1223 m	1236 m	ν(CN), δ(CH)
-	1142 m	1155 vs	1153 m	1147 m	1157 m	1159 m	ν(CC), ν(CN), δ(CH)
920 m	929 s	942 s	956 s	941 m	930 m	933 m	δ(CH), δ(imidazole ring)
855 m	-	875 w	875 w	862 m	865 m	856 m	π(CH), δ(imidazole ring)
-	750 s	756 vs	751 vs	756 m	755 m	754 m	π(CH)
-	615 m	627 w	625 w	623 m	627 m	624 m	π(NH)

HIm—imidazole; 2-MeIm—2-methylimidazole; 2-EtIm—2-ethylimidazole; NaMacr—sodium metacrylate; (**1**)—[Co(Macr)_2_(Im)_2_]; (**2**)—[Co(Macr)_2_(2-MeIm)_2_]; (**3**)—[Co(Macr)_2_(2-EtIm)_2_]; ν—stretching; δ—in plane bending; π—out of plane bending; vs.—very strong; s—strong; m—medium; w—weak (absorption band intensity).

**Table 4 molecules-26-00055-t004:** Absorption maxima in UV-Vis-NIR spectra of complexes (**1**)–(**3**).

Compound	Absorption Maxima		Assignments
	λ [nm]	ῦ [cm^−1^]	
[Co(Macr)_2_(Im)_2_] (**1**)	260	38,460	π → π*
	525	19,050	^4^A′_2_ → ^4^E′′ (P)
	565	17,700	^4^A′_2_ → ^4^A′_2_ (P)
	865 1095	11,560 9130	^4^A′_2_ → ^4^E′
	1385	7220	^4^A′_2_ → ^4^E″
	1925	5194	^4^A′_2_ → ^4^A″_1_, ^4^A″_2_
[Co(Macr)_2_(2-MeIm)_2_] (**2**)	265	37,735	π → π*
	535	18,520	^4^A′_2_ → ^4^E′′ (P)
	570	17,540	^4^A′_2_ → ^4^A′_2_ (P)
	880 1135	11,360 8810	^4^A′_2_ → ^4^E′
	1395	7170	^4^A′_2_ → ^4^E″
	1840	5435	^4^A′_2_ → ^4^A″_1_, ^4^A″_2_
[Co(Macr)_2_(2-EtIm)_2_] (**3**)	260	38,460	π → π*
	545	18,350	^4^A_2_ → ^4^E (P)
	585	17,090	^4^A_2_ → ^4^T_1_(P) T_d_
	910	11,000	^4^A_2_ → ^4^B_1_
	1190	8400	^4^A_2_ → ^4^T_1_(F) T_d_
	1395	7000	^4^A_2_ → ^4^E
	1920	5200	^4^A_2_ → ^4^B_2_

**Table 5 molecules-26-00055-t005:** Thermal decomposition data (in air flow) for complexes (**1**)–(**3**).

Complex	Step	Thermal Effect	Temp./°C	Δ*m_exp_*_/_%	Δ*m_calc_*_/_%	Process
[Co(Macr)_2_(Im)_2_] (**1**)	1.	Endothermic	135	-	-	melting
	2.	Miscellaneous	220–480	36.10	37.27	imidazole elimination
	3.	Exothermic	480–900	40.58	40.77	methacrylate oxidative degradation
	Residue (Co_3_O_4_)			23.32	21.96	
[Co(Macr)_2_(2-MeIm)_2_] (**2**)	1.	Endothermic	119	-	-	melting
	2.	Miscellaneous	270–490	42.33	41.75	2-methylimidazole elimination
	3.	Exothermic	490–790	36.67	37.86	methacrylate oxidative degradation
	Residue (Co_3_O_4_)			21.00	20.39	
[Co(Macr)_2_(2-EtIm)_2_] (**3**)	1.	Endothermic	135	-	-	melting
	2.	Miscellaneous	220–480	45.8	45.62	2-ethylimidazole elimination
	3.	Exothermic	480–900	35.06	35.33	methacrylate oxidative degradation
	Residue (Co_3_O_4_)			19.14	19.05	

**Table 6 molecules-26-00055-t006:** Minimal inhibitory concentration (MIC)/minimal biofilm eradication concentration (MBEC) values (μg/mL) of the ligands, cobalt (II) methacrylate, and complexes (**1**)–(**3**).

Microbial Strains	Co(Macr)_2_		HIm		2-MeIm		2-EtIm		(1)		(2)		(3)	
	MIC	MBEC	MIC	MBEC	MIC	MBEC	MIC	MBEC	MIC	MBEC	MIC	MBEC	MIC	MBEC
*E. coli* ATCC 8739	62.5	>500	125	>500	125	>500	62.5	>500	31.2	31.2	31.2	31.2	125	62.5
*P. aeruginosa* ATCC 1671	250	500	62.5	500	31.2	500	31.2	500	62.5	15.6	31.2	15.6	31.2	31.2
*S. aureus* ATCC 6538	125	500	>500	500	>500	500	>500	500	15.6	15.6	15.6	15.6	15.6	15.6
*E. faecalis* ATCC 29212	250	500	>500	500	>500	500	500	500	31.2	31.2	62.5	62.5	62.5	62.5
*C. albicans* ATCC 26790	250	>500	500	>500	250	>500	500	>500	7.8	>500	7.8	>500	15.6	>500

**Table 7 molecules-26-00055-t007:** Effect of co-imidazole compounds on cell cycle phases of HeLa, HCT-8, and MG63 cells. The results are presented as % of cells found in different stages of the cellular cycle.

	(1)		(2)		(3)		Control
HeLa							
	50 µg/mL	100 µg/mL	50 µg/mL	100 µg/mL	50 µg/mL	100 µg/mL	
G1	58.29	30.98	59.48	28.43	59.65	22.74	58.84
S	24.95	42.86	33.96	36.2	23	40.48	25.97
G2/M	8.36	15.95	7.73	10.59	8.14	13.64	8.51
HCT-8							
G1	71.50	52.24	62.84	66.98	70.59	60.75	88.70
S	8.02	26.90	10.85	21.20	7.75	23.12	8.66
G2/M	6.86	6.82	8.24	14.92	3.33	17.18	3.81
MG63							
G1	34.19	26.64	57.55	15.51	34.97	21.2	81.49
S	25.29	19.38	19.86	24.93	26.68	25.67	10.75
G2/M	30.48	51.44	17.58	51.71	30.25	52.66	6.35

## Data Availability

The data presented in this study are available on request from the corresponding author.

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
