# Peer review of "New Cobalt (II) Complexes with Imidazole Derivatives: Antimicrobial Efficiency against Planktonic and Adherent Microbes and In Vitro Cytotoxicity Features"

_molecules, 2020, doi:10.3390/molecules26010055_

Round 1

Reviewer 1 Report

The paper can be published in its current form.

Author Response

We solved the references problem.

Reviewer 2 Report

Authors have made progresses compare to the previous submission.

References can not be seen in the text in several places. These need to be corrected.

Author Response

The problem with references arose because we used cross-reference. We solved that.

This manuscript is a resubmission of an earlier submission. The following is a list of the peer review reports and author responses from that submission.

Round 1

Reviewer 1 Report

The paper is in my opinion scientifically sound and as such it is publishable, however I do not find it interesting for a wider readership. The obtained species are very close both in their structure as well properties. They also show pretty low biological activity. In some fragments e.g. IR spectra, the discussion of presented results should be expanded. If it comes to the antimicrobial activity, results for an appropriate negative as well as positive controls should definitely be given. Claiming that the antifungal activity of the species is very good is not justified and should be removed from abstract. It would be also interesting to know the stability of reported species in solution, especially that their biological activity has been determined in solution. Below please find some minor issues:

  • Please explain the MBEC abreviation in Abstract where it is mentioned.
  • Line 46: “are predicting” is incorrect
  • Line 79: “polyinuclear” to be corrected
  • I recommend moving “as raw materials” from line 92 to line 93 at the end of the sentence
  • Line 133: “Co3”?
  • Line 188: “growth strate”?
  • Lines 202-206: The sentence is far too long and should be divided into shorter to be understandable
  • Line 329: “un/- treated”?
  • Please correct formatting of refs:2, 6, 22.

Reviewer 2 Report

This study is aimed to report the synthesis, structural characterization and results related to biological activity for 85 cobalt (II) complexes with mixed ligands, methacrylate anion and imidazole derivatives. The study is well designed and prepared in a sufficient level. The study seems original and contains new results that can be considered for publication after suitable revisions.

Suggestions and comments:

L20: IR- give in full, not only the abbreviated form

L26 and 28: MIC - give in full, not only the abbreviated form

L28: MBEC - give in full, not only the abbreviated form

L148: Tables should be self explanatory: IR - give in full, not only the abbreviated form

L182: Figures should be self explanatory: TG, DTG and DTA - give in full, not only the abbreviated form

L189: C. albicans – it was already introduced

L194: S. aureus– it was already introduced

L212: Tables should be self explanatory: MIC/MBEC - give in full, not only the abbreviated form. Also give the full names of the microorganisms.

L220: Figures are too small.

L240: Figure 8: No title for x axis. Give the meaning of 1,2 and 3 in the figure.

L348: Give the conclusions in points.

L391: space missing ’ Anorg.Allg.Chem.’

Reviewer 3 Report

The presented manuscript has a interesting and good chemical part, but the biological section is poorly developed and poorly conducted. It renders the article unfit for publication.

The biological assays are not correlated between them. It looks like a sum of assays put together without any clear purpose. If the objective was to develop an antimicrobial agent, why also test the antitumor effect?

The experimental protocols have no real logic. Why the Cell viability or Cytotoxicity (it is no clear which was measured) was done after 72 hours? It seems a long interval. Usually it is 24 or 48 hours. Also, why the “semi-quantitative real-time RT-PCR” was done after 24 hours? And why use the dose of 100 µg/mL? The authors declare that the cell cycle analysis was done at 100 µg/mL, but in figure 7 there are both 50 µg/mL and 100 µg/mL.

The section 222-252 is added in my opinion without any logic. They measured the expressions of some protein targets without any clear motive. They seem random. There are other protein more important that could indicate better the mechanism of these compounds then these proteins.

Authors should have demonstrated that these compounds have no effect on the normal cells. All the essays have no positive standard. They should test a known compound together with these 3 compounds in order to assess their effect. It is wrong to use mg/mL as expression of concentration. The authors should transform them to molar concentrations. How where the IC50 values calculated? How many points? There are no statistical data? Any measure of deviation from the average, the confidence interval? How many experiments were performed? Just one?

All these data is missing making the value of the biological assay very poor.

The figure 7 is useless in my opinion, adding no important data. A simple table could be provided.

I recommend the authors to perform major changes on their paper. If the chemistry part has logic and seems solid, all the biological section seems random. It adds more questions than provides answers. I recommend the authors to remove all the “antitumor” section and develop better the antimicrobial section by testing standard compounds along the complexes. Also they should test alone the metal and the ligands at the same molar concentrations as those in the complexes.

I think the Acknowledgments section is wrong, the projects mentioned there are not by far related to the content of the article “…Professional Training in Archaeological Sciences…” and “…New technologies for preservation, conservation, recovering and restauration of the cultural heritage…” I advise the authors to check and correct them.